# Rate pressure product as a novel predictor of long-term adverse outcomes in patients after percutaneous coronary intervention: a retrospective cohort study

Zhi-Hui Jiang, Abudula Aierken, Ting-Ting Wu, Ying-Ying Zheng, Yi-Tong Ma, Xiang Xie

Department of Cardiology, First Affiliated Hospital of Xinjiang Medical University, Urumqi, Xinjiang, China

**Correspondence to**
Dr Xiang Xie;
xiangxie999@sina.com

## ABSTRACT

**Objectives** Previous studies have suggested that heart rate and blood pressure play important roles in the development of adverse outcomes in patients with coronary artery disease (CAD) who underwent percutaneous coronary intervention (PCI). However, the relationship between the rate pressure product (RPP) and long-term outcomes has rarely been investigated. This study investigated the effects of RPP on the clinical outcomes of patients with CAD who underwent PCI.

**Design** In this study, a total of 6015 patients with CAD were enrolled. All patients were from the CORFCHD-PCI (Clinical Outcomes and Risk Factors of Patients with Coronary Heart Disease after PCI) Study. They were divided into two groups according to RPP (RPP <10 269, n=4018 and RPP ≥10 269, n=1997). In addition, the median follow-up time was 32 months.

**Participants** Data was obtained from 6050 patients with CAD who underwent PCI at the First Affiliated Hospital of Xinjiang Medical University from January 2008 to December 2016.

**Primary and secondary outcome measures** The primary endpoint was long-term mortality, including all-cause mortality (ACM) and cardiac mortality (CM). The secondary endpoints were major adverse cardiovascular events (MACEs) and major adverse cardiovascular and cerebrovascular events (MACCEs).

**Results** We found that there were significant differences between the two groups in the incidence of ACM, CM, MACCEs and MACEs (all p<0.05). Among the patients with CAD having ACM, CM, MACCEs and MACEs, the mean survival time of the low-value group was significantly higher than that of the high-value group. Multivariate Cox regression analyses showed that RPP was an independent predictor for ACM (HR=1.605, 95% CI: 1.215–2.120, p=0.001), CM (HR=1.733, 95% CI: 1.267–2.369, p=0.001), MACCEs (HR=1.271, 95% CI: 1.063–1.518, p=0.008) and MACEs (HR=1.315, 95% CI: 1.092–1.584, p=0.004) in patients with stable CAD. On the other hand, there was no significant correlation between the RPP and the adverse outcomes in patients with acute coronary syndrome.

**Conclusion** In summary, RPP is an independent predictor of long-term prognosis in patients with CAD who underwent PCI. A higher baseline RPP before PCI increased the risk of adverse outcomes. Compared with heart rate and blood pressure alone, RPP has a higher predictive value for adverse clinical outcomes.

## STRENGTHS AND LIMITATIONS OF THIS STUDY

⇒ The present study was a single-centre retrospective cohort study with a larger sample number and a long follow-up time, which can improve the scientific nature of the results.
⇒ The present study only collected heart rate and blood pressure data at the first hospitalisation, thus lacking information on dynamic changes in these variables.
⇒ The present study did not rule out the effect of the patient's current medication such as antihypertensive drugs on blood pressure levels.
⇒ Follow-up was mainly based on telephone contacts and the method of measurement was mainly manual, which may have impacted the results.

## INTRODUCTION

Coronary atherosclerotic heart disease refers to the occurrence of atherosclerosis in the coronary arteries, which narrows or obstructs the arteries, causing local myocardial ischaemia, hypoxia and even necrosis, resulting in changes in heart structure and loss of function. Together with coronary artery spasms, they are collectively called coronary artery disease (CAD).[1] CAD is one of the main causes of death and disability in China, and is a serious threat to health. The WHO classifies CAD into five clinical types: silent myocardial ischaemia (latent CAD), angina pectoris, myocardial infarction, ischaemia heart failure (ischaemic heart disease) and sudden cardiac death. The related risk factors include sex, age, smoking, insufficient exercise, obesity, diabetes, dyslipidaemia and

metabolic disorders, hypertension, family history of CAD and others.[2–5] At present, percutaneous coronary intervention (PCI) is one of the most important methods for the treatment of CAD, and can significantly reduce the mortality and risk of reischaemic attacks.[6–8] As an established measure of myocardial load, rate pressure product (RPP) may provide a more reliable assessment of cardiac workload.[9] Previous studies have shown that RPP can be used as an auxiliary reference index for CAD, which is negatively correlated with the degree of CAD and the number of diseased branches and negatively correlated with age and sex.[10] RPP indirectly reflects myocardial blood supply and oxygen consumption. Cook *et al*[11] found that PCI can increase myocardial workload. The change observed immediately following PCI was caused by the abolition of stenosis resistance. Yazdani *et al*[12] found that compared with systolic blood pressure (SBP), diastolic blood pressure (DBP) and heart rate, RPP was a stronger predictor of all-cause mortality (ACM) and cardiac mortality (CM) in the general population and in patients with CAD, especially in the cohort of patients at high risk of cardiovascular disease. Interestingly, RPP also predicted CM in patients with heart failure with preserved ejection fraction in a stronger way, while SBP and heart rate alone showed no significant association. Furthermore, in patients with three-vessel CAD, RPP was a stronger predictor of CM than SBP or heart rate alone. Therefore, especially in patients with severe CAD and heart failure, the cardiac workload given as RPP should be considered in clinical practice.[12] However, they did not clarify the predictability of RPP for the prognosis of patients with CAD who underwent PCI. The RPP was defined as the product of SBP (mm Hg) and heart rate (beats/min), because it combined the information of the two indicators, and both indicators were related to the prognosis of patients with CAD. Therefore, we speculated that RPP may affect the long-term prognosis of patients with CAD after PCI. Compared with heart rate and SBP, RPP may have a higher predictive value for adverse clinical outcomes.

In this research, 6015 patients with CAD were included to investigate the relationship between the RPP and the clinical outcomes.

## METHODS
### Study design and population
In this study, 6050 patients with CAD who were hospitalised at the First Affiliated Hospital of Xinjiang Medical University from January 2008 to December 2016 were evaluated. Regarding the inclusion criteria, coronary angiography confirmed that at least one major coronary artery showed stenosis ≥70%, including the left main coronary artery, left anterior descending artery, left circumflex artery and right coronary artery, or other more important branches. All patients underwent PCI and received at least one stent via implantation. Regarding the exclusion criteria, patients with the presence of serious heart

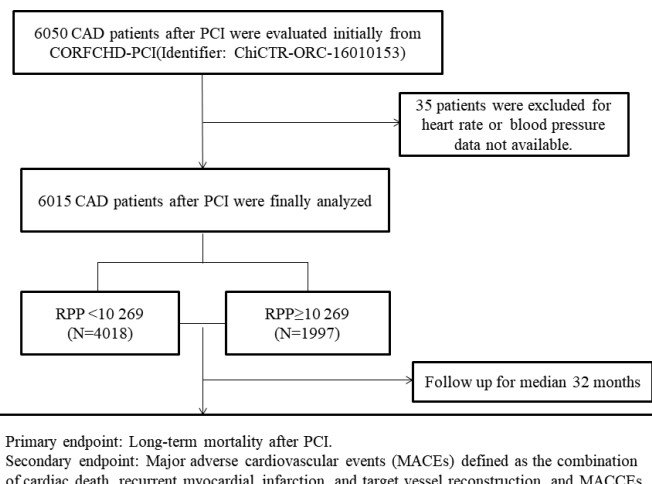

**Figure 1** The participant flow chart. CAD, coronary artery disease; CORFCHD-PCI, Clinical Outcomes and Risk Factors of Patients with Coronary Heart Disease after PCI; MACCEs, major adverse cardiovascular and cerebrovascular events; PCI, percutaneous coronary intervention; RPP, rate pressure product.

failure, rheumatic heart disease, valvular heart disease, congenital heart disease, pulmonary heart disease, serious dysfunction of the liver or kidney, acute infections, malignant tumour, alcohol abuse or blood disease were excluded. To investigate the relation between the RPP and the outcomes in patients with CAD who underwent PCI, a total of 6050 patients were evaluated initially according to the inclusion and exclusion criteria. Thirty-five patients were excluded due to incomplete follow-up data. Finally, 6015 patients were enrolled in this study, including 2029 patients with acute coronary syndrome (ACS) and 3986 patients with stable CAD. All patients with CAD were from the Clinical Outcomes and Risk Factors of Patients with Coronary Heart Disease after PCI (CORFCHD-PCI) Study. The study is a large single-centre retrospective cohort study based on medical records and the follow-up information collected at the First Affiliated Hospital of Xinjiang Medical University. The details of the design have been registered at http://www.chictr.org.cn (ChiCTR-ORC-16010153). Figure 1 shows the flow chart of the inclusion and exclusion criteria for the selection of 6050 patients with CAD after PCI.

### Patient and public involvement
No patient involved.

### Data collection
The cardiovascular-related risk factors, clinical data, laboratory data and interventional therapy data of all patients were collected retrospectively, including: age, sex, heart rate, blood pressure (BP), smoking status, alcohol consumption, history of diagnosed diabetes, history of hypertension, family history of CAD, history of medication, blood urea nitrogen (BUN), serum creatinine (Cr), uric acid (UA), glucose (GLU), triglycerides

(TGs), total cholesterol (TC), low-density lipoprotein cholesterol (LDL-C), high-density lipoprotein cholesterol (HDL-C) and left ventricular ejection fractions (LVEF). Heart rate and BP at admission were measured accurately according to a highly standardised protocol at the first medical contact by experienced physicians using calibrated manual mercury sphygmomanometers with an appropriately sized cuff. One sphygmomanometer was placed on the right arm, and after at least 5 min of rest in the sitting position. Two heart rate and BP measurements were obtained and taking the average, allowing for a 1 min interval between measurements. All laboratory tests were carried out in the Medical Laboratory Centre of the First Affiliated Hospital of Xinjiang Medical University.

## Clinical diagnosis
The diagnostic criteria for diabetes mellitus were a clear history of diabetes, the use of hypoglycaemic agents, fasting plasma glucose ≥7.0 mmol/L, 2-hour plasma glucose ≥11.1 mmol/L during oral glucose tolerance test, A1C ≥6.5% (48 mmol/mol) or a random plasma glucose ≥11.1 mmol/L in a patient with classic symptoms of hyperglycaemia or hyperglycaemic crisis.[13] Hypertension was diagnosed when SBP was ≥140 mm Hg and/or DBP was ≥90 mm Hg following repeated examination or treatment with antihypertensive drugs.[14]

## Endpoints
The primary endpoint was long-term mortality, including long-term ACM and CM. The secondary endpoints were major adverse cardiovascular events (MACEs) and major adverse cardiovascular and cerebrovascular events (MACCEs). MACEs referred to the composite endpoint of cardiac death, recurrent myocardial infarction and target vessel reconstruction. MACCEs was defined as MACEs plus non-fatal stroke.[15]

## Follow-up
In this study, 6050 patients with CAD started receiving follow-up when they underwent PCI, and regular follow-up was performed at the end of 1 month, 3 months, 6 months, 1 year, 3 years and 5 years after discharge. Outpatient review and telephone were used to follow up until the endpoint events occurred or the study was concluded. The median follow-up time was 32 months, and the longest follow-up time was 10 years.

## Statistical analyses
All statistical analyses were performed using SPSS V.26.0. Continuous data are presented as the mean±SD ($\bar{x}$±s), and categorical variables are presented as frequencies and percentages. RPP is a continuous numerical variable. Because there is no established optimal threshold for RPP, receiver operating characteristic (ROC) curve analysis was performed to assess the predictive ability of RPP in clinical outcomes. The point where the sensitivity and specificity were maximised was determined as the best cut-off point. Two samples t-tests or $X^2$ tests were used to compare the clinical data of patients in different groups

and the characteristics of adverse outcomes. Kaplan-Meier analysis was used to compare the cumulative incidence rates of long-term adverse outcomes and the log-rank test was used to compare between groups. Multivariate Cox regression analysis was performed to evaluate the predictive value of RPP for clinical outcomes. The fully adjusted models for RPP included age, hypertension, diabetes mellitus, sex, smoking, alcohol consumption, LDL-C, TC, GLU, BUN, post-dilatation, the number of vascular lesions, chronic total occlusion lesions (CTO), multivessel lesions (MLs) and aspirin. HRs and 95% CIs were also calculated. A p value of <0.05 was considered statistically significant.

## RESULTS
### Baseline data
Among the 6015 patients included in this study, 74.3% were male, 40.0% were smokers, 29.3% were drinkers, 24.1% had diabetes mellitus, 42.3% had hypertension and the mean age of the patients was 59.48 years. During the follow-up period, a history of using calcium antagonists accounted for 11.5%, and patients using β-receptor blockers, ACE inhibitor (ACEI)/angiotensin receptor blocker (ARB) drugs, statins, aspirin and clopidogrel accounted for 40.2%, 22.7%, 53.9%, 66.9% and 30.4%, respectively. During the 10-year follow-up, 560 (9.3%) deaths occurred, of which 251 (4.2%) patients died due to cardiovascular disease. MACCEs occurred in 859 (14.3%) patients, including 782 (13.0%) MACEs and 77 (1.28%) non-fatal strokes.

In this study, the ROC curve analysis showed that an RPP value of 10 269 was the cut-off with the highest sensitivity and specificity in terms of prognostic significance. The population clinical characteristics were grouped by dichotomy of RPP, which was based on the ROC cut-off. RPP <10 269 was considered the low-value group, and RPP ≥10 269 was considered the high-value group. There were 4018 patients with CAD in the low-value group and 1997 patients with CAD in the high-value group. As shown in table 1, we found that there were significant differences in many variables between the two groups (all p<0.05), specifically, age, BUN, GLU, TC and LDL-C in the high-value group were higher than those in the low-value group. Sex, smoking, alcohol consumption, diabetes mellitus, hypertension, and the therapy of statins and aspirin in the high-value group were lower than those in the low-value group. We did not find statistically significant differences between the two groups in regard to Cr, UA, TG, HDL-C, LVEF, or the therapy of calcium channel blocker, β-receptor blocker, ACEI or ARB, as well as clopidogrel (all p>0.05). In the patients with ACS, we found that diabetes mellitus, hypertension, GLU, TG, TC and clopidogrel were significantly different between the two groups. In the patients with stable CAD, we found that age, sex, smoking, alcohol drinking, diabetes, hypertension, BUN, UA, GLU, and the therapy of aspirin and clopidogrel were significantly different between the two groups (all p<0.05) (online supplemental table 1). For PCI procedures themselves,

**Table 1** Characteristics of participants of the two groups

| Variables | Total | | X² or t | P value |
|---|---|---|---|---|
| | RPP <10 269 (n=4018) | RPP ≥10 269 (n=1997) | | |
| Age, years | 58.92±10.90 | 60.62±10.61 | −5.751 | **<0.001** |
| Male, n (%) | 3057 (76.1) | 1415 (70.9) | 19.105 | **<0.001** |
| Smoking, n (%) | 1686 (42.0) | 718 (36.0) | 20.064 | **<0.001** |
| Alcohol drinking, n (%) | 1217 (30.3) | 544 (27.2) | 5.985 | **0.014** |
| Diabetes, n (%) | 847 (21.1) | 600 (30.0) | 58.684 | **<0.001** |
| Hypertension, n (%) | 1478 (36.8) | 1067 (53.4) | 151.429 | **<0.001** |
| BUN, mmol/L | 5.48±1.65 | 5.59±1.72 | −2.271 | **0.023** |
| Cr, µmol/L | 75.78±19.55 | 76.03±21.97 | −0.447 | 0.655 |
| UA, mmol/L | 323.96±89.56 | 322.21±91.03 | 0.697 | 0.486 |
| GLU, mmol/L | 6.41±2.99 | 6.92±3.39 | −5.888 | **<0.001** |
| TG, mmol/L | 1.88±1.23 | 1.94±1.35 | −1.751 | 0.080 |
| TC, mmol/L | 3.94±1.10 | 4.01±1.13 | −2.305 | **0.021** |
| LDL-C, mmol/L | 2.44±0.90 | 2.51±0.95 | −2.689 | **0.007** |
| HDL-C, mmol/L | 1.02±0.46 | 1.02±0.52 | −0.196 | 0.845 |
| CCB, n (%) | 462 (11.6) | 227 (11.4) | 0.022 | 0.883 |
| β-blocker, n (%) | 1631 (40.8) | 785 (39.5) | 0.930 | 0.335 |
| ACEI or ARB, n (%) | 894 (22.4) | 472 (23.8) | 1.525 | 0.217 |
| Statins, n (%) | 2207 (55.5) | 1035 (52.2) | 5.480 | **0.019** |
| LVEF, % | 61.10±7.09 | 61.00±6.97 | 0.454 | 0.650 |
| Aspirin, n (%) | 2748 (68.9) | 1275 (64.2) | 13.062 | **<0.001** |
| Clopidogrel, n (%) | 1228 (30.8) | 599 (30.2) | 0.237 | **0.626** |

The boldfaced values indicate p<0.05.
ACEI, angiotensin-converting enzyme inhibitor; ARB, angiotensin receptor blocker; BUN, blood urea nitrogen; CCB, calcium channel blocker; Cr, creatinine; GLU, glucose; HDL-C, high-density lipoprotein cholesterol; LDL-C, low-density lipoprotein cholesterol; LVEF, left ventricular ejection fraction; RPP, rate pressure product; TC, total cholesterol; TG, triglyceride; UA, uric acid.

several variables were significantly different between these two groups, especially in the patients with ACS. These variables included post-dilatation pressure, CTO, ML as well as the number of vascular lesions (all p<0.05) (online supplemental table 2).

### Clinical outcomes

A comparison of clinical outcomes between the two groups of patients with CAD is shown in table 2. For the primary endpoints, the incidence rate of all-cause death was 6.4% in the high-value group and 4.5% in the low-value group. The difference was significant (p=0.002); also, the incidence rate of cardiac death between the two groups showed a significant difference (5.3% vs 3.6%, p=0.002). For the secondary endpoints, we found that there were significant differences between the two groups in the incidence of MACCEs (15.8% vs 13.5%, p=0.020) and MACEs (14.4% vs 12.3%, p=0.020).

Subgroup analysis suggested that there were significant differences in the incidence of ACM (6.9% vs 4.8%, p=0.004), CM (5.7% vs 3.7%, p=0.003) and MACEs (14.4% vs 12.2%, p=0.040) between the high-value group and the low-value group for the patients with stable CAD.

**Table 2** Outcomes comparison between groups

| Outcomes | Total | | X² | P value |
|---|---|---|---|---|
| | RPP <10 269 (n=4018) | RPP ≥10 269 (n=1997) | | |
| ACM, n (%) | 181 (4.5) | 128 (6.4) | 9.933 | **0.002** |
| CM, n (%) | 145 (3.6) | 106 (5.3) | 9.632 | **0.002** |
| MACCEs, n (%) | 544 (13.5) | 315 (15.8) | 5.441 | **0.020** |
| MACEs, n (%) | 494 (12.3) | 288 (14.4) | 5.336 | **0.021** |

The boldfaced values indicate p<0.05.
ACM, all-cause mortality; CM, cardiac mortality; MACCEs, major adverse cardiovascular and cerebrovascular events; MACEs, major adverse cardiovascular events; RPP, rate pressure product.

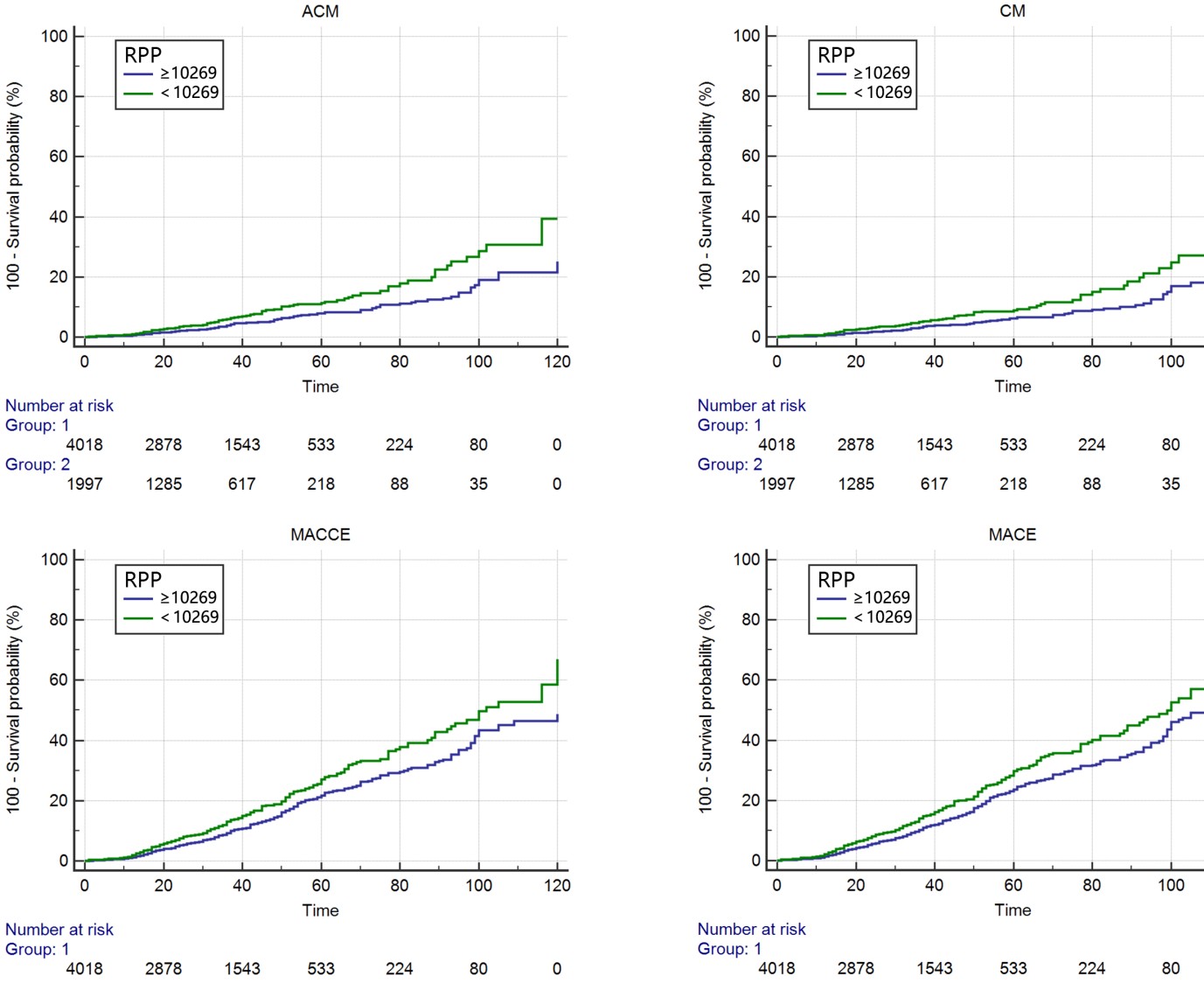

**Figure 2** Cumulative Kaplan-Meier estimates of the time to the first adjudicated occurrence of primary endpoints and secondary endpoints: group 1 is the low-value group with RPP <10269, and group 2 is the high-value group with RPP ≥10269. ACM, all-cause mortality; CM, cardiac mortality; MACCEs, major adverse cardiovascular and cerebrovascular events; MACEs, major adverse cardiovascular events; RPP, rate pressure product.

For the patients with ACS, we found no significant difference in adverse outcomes (all p>0.05) (online supplemental table 3).

In this study, baseline RPP before PCI was used as an independent variable, and ACM, CM, MACEs and MACCEs were used as the endpoints. Kaplan-Meier analysis was performed, and the log-rank test was used for comparisons between groups. Among the patients with CAD having ACM, CM, MACCEs and MACEs, the mean survival time of the low-value group was significantly higher than that of the high-value group (online supplemental table 4). As shown in figure 2, with the extension of the follow-up time, the cumulative survival rate of the patients showed a downward trend, and the decreasing rate of the high-value group was more significant than that of the low-value group. While the cumulative risk showed an upward trend, the increasing rate of the high-value group was significantly higher than that of the

low-value group. The above differences were statistically significant (all p<0.05).

Tables 3–6 show select variables based on the aforementioned analysis. The results of collinearity analysis showed that there was no significant collinearity among age, hypertension, diabetes mellitus, sex, GLU, BUN, LDL-C, TC, smoking, alcohol consumption, post-dilatation, the number of vascular lesions, CTO, ML, aspirin and RPP. Multivariate Cox regression analysis was performed to assess the prognostic value of the RPP and adverse outcomes after adjusting for the influence of age, hypertension, diabetes mellitus, sex, GLU, BUN, LDL-C, TC, smoking, alcohol consumption and other confounding factors. In total, the results showed that higher preoperative RPP value was a risk factor for clinical outcomes in patients with CAD who underwent PCI. The respective risks of ACM (HR=1.598, 95% CI: 1.262–2.024, p<0.001), CM (HR=1.678, 95% CI: 1.289–2.184, p<0.001), MACCEs

**Table 3** Multivariable Cox regression analysis for ACM

| Variables | Total | | | | |
| --- | --- | --- | --- | --- | --- |
| | B | SE | Wald | P value | HR (95% CI) |
| Age | 0.026 | 0.006 | 19.314 | **<0.001** | 1.027 (1.015–1.039) |
| Male | −0.010 | 0.149 | 0.005 | 0.944 | 0.990 (0.739–1.325) |
| Smoking | −0.035 | 0.153 | 0.053 | 0.817 | 0.965 (0.715–1.303) |
| Alcohol drinking | −0.017 | 0.162 | 0.011 | 0.915 | 0.983 (0.715–1.351) |
| Diabetes | 0.032 | 0.149 | 0.047 | 0.828 | 1.033 (0.772–1.382) |
| Hypertension | 0.186 | 0.123 | 2.282 | 0.131 | 1.205 (0.946–1.534) |
| GLU | −0.012 | 0.020 | 0.361 | 0.548 | 0.988 (0.949–1.028) |
| TC | 0.088 | 0.087 | 1.027 | 0.311 | 1.092 (0.921–1.296) |
| BUN | 0.079 | 0.032 | 6.088 | **0.014** | 1.082 (1.016–1.153) |
| LDL-C | −0.172 | 0.109 | 2.480 | 0.115 | 0.842 (0.679–1.043) |
| Post-dilatation | 0.132 | 0.124 | 1.140 | 0.286 | 1.141 (0.896–1.454) |
| Number of vascular lesions | 0.180 | 0.143 | 1.575 | 0.209 | 1.197 (0.904–1.584) |
| CTO | 0.409 | 0.134 | 9.272 | **0.002** | 1.505 (1.157–1.957) |
| MLs | 0.032 | 0.262 | 0.015 | 0.903 | 1.032 (0.618–1.724) |
| Aspirin | −2.141 | 0.194 | 121.798 | **<0.001** | 0.118 (0.080–0.172) |
| RPP | 0.469 | 0.121 | 15.098 | **<0.001** | 1.598 (1.262–2.024) |

The boldfaced values indicate p<0.05.
ACM, all-cause mortality; BUN, blood urea nitrogen; CTO, chronic total occlusion lesions; GLU, glucose; LDL-C, low-density lipoprotein cholesterol; MLs, multivessel lesions; RPP, rate pressure product; TC, total cholesterol.

**Table 4** Multivariable Cox regression analysis for CM

| Variables | Total | | | | |
| --- | --- | --- | --- | --- | --- |
| | B | SE | Wald | P value | HR (95% CI) |
| Age | 0.018 | 0.007 | 7.267 | **0.007** | 1.018 (1.005–1.031) |
| Male | −0.013 | 0.166 | 0.006 | 0.937 | 0.987 (0.712–1.368) |
| Smoking | −0.157 | 0.173 | 0.824 | 0.364 | 0.855 (0.609–1.199) |
| Alcohol drinking | 0.049 | 0.181 | 0.072 | 0.788 | 1.050 (0.736–1.498) |
| Diabetes | 0.176 | 0.164 | 1.151 | 0.283 | 1.192 (0.865–1.643) |
| Hypertension | 0.120 | 0.139 | 0.748 | 0.387 | 1.127 (0.859–1.479) |
| GLU | −0.037 | 0.025 | 2.212 | 0.137 | 0.964 (0.919–1.012) |
| TC | 0.144 | 0.094 | 2.343 | 0.126 | 1.155 (0.960–1.388) |
| BUN | 0.109 | 0.035 | 9.712 | **0.002** | 1.115 (1.041–1.194) |
| LDL-C | −0.209 | 0.119 | 3.088 | 0.079 | 0.811 (0.642–1.024) |
| Post-dilatation | 0.135 | 0.138 | 0.956 | 0.328 | 1.145 (0.873–1.501) |
| Number of vascular lesions | 0.277 | 0.161 | 2.955 | 0.086 | 1.319 (0.962–1.810) |
| CTO | 0.466 | 0.148 | 9.861 | **0.002** | 1.594 (1.191–2.132) |
| MLs | −0.105 | 0.298 | 0.124 | 0.724 | 0.900 (0.502–1.614) |
| Aspirin | −2.074 | 0.209 | 98.081 | **<0.001** | 0.126 (0.083–0.189) |
| RPP | 0.518 | 0.134 | 14.812 | **<0.001** | 1.678 (1.289–2.184) |

The boldfaced values indicate p<0.05.
BUN, blood urea nitrogen; CM, cardiac mortality; CTO, chronic total occlusion lesions; GLU, glucose; LDL-C, low-density lipoprotein cholesterol; MLs, multivessel lesions; RPP, rate pressure product; TC, total cholesterol.

**Table 5** Multivariable Cox regression analysis for MACCEs

| Variables | Total | | | | |
| --- | --- | --- | --- | --- | --- |
| | B | SE | Wald | P value | HR (95% CI) |
| Age | −0.002 | 0.004 | 0.332 | 0.565 | 0.998 (0.991–1.005) |
| Male | −0.144 | 0.092 | 2.454 | 0.117 | 0.866 (0.723–1.037) |
| Smoking | −0.217 | 0.091 | 5.704 | **0.017** | 0.805 (0.673–0.962) |
| Alcohol drinking | −0.048 | 0.096 | 0.250 | 0.617 | 0.953 (0.790–0.151) |
| Diabetes | 0.176 | 0.086 | 4.123 | **0.042** | 1.192 (1.006–1.412) |
| Hypertension | 0.314 | 0.074 | 18.238 | **<0.001** | 1.369 (1.185–1.581) |
| GLU | −0.005 | 0.012 | 0.143 | 0.705 | 0.995 (0.972–1.019) |
| TC | 0.019 | 0.054 | 0.121 | 0.728 | 1.019 (0.916–1.134) |
| BUN | 0.055 | 0.020 | 7.464 | **0.006** | 1.057 (1.016–1.099) |
| LDL-C | −0.112 | 0.067 | 2.746 | 0.097 | 0.894 (0.784–1.021) |
| Post-dilatation | 0.040 | 0.073 | 0.292 | 0.589 | 1.040 (0.901–1.201) |
| Number of vascular lesions | 0.185 | 0.087 | 4.513 | **0.034** | 1.204 (1.014–1.428) |
| CTO | 0.200 | 0.084 | 5.754 | **0.016** | 1.222 (1.037–1.439) |
| MLs | −0.032 | 0.157 | 0.042 | 0.838 | 0.968 (0.712–1.317) |
| Aspirin | −0.493 | 0.082 | 36.279 | **<0.001** | 0.611 (0.520–0.717) |
| RPP | 0.234 | 0.074 | 9.840 | **0.002** | 1.263 (1.092–1.462) |

The boldfaced values indicate p<0.05.
BUN, blood urea nitrogen; CTO, chronic total occlusion lesions; GLU, glucose; LDL-C, low-density lipoprotein cholesterol; MACCEs, major adverse cardiovascular and cerebrovascular events; MLs, multivessel lesions; RPP, rate pressure product; TC, total cholesterol.

**Table 6** Multivariable Cox regression analysis for MACEs

| Variables | Total | | | | |
| --- | --- | --- | --- | --- | --- |
| | B | SE | Wald | P value | HR (95% CI) |
| Age | −0.004 | 0.004 | 1.34 | 0.247 | 0.996 (0.989 to −1.003) |
| Male | −0.115 | 0.097 | 1.391 | 0.238 | 0.892 (0.737 to −1.079) |
| Smoking | −0.156 | 0.095 | 2.721 | 0.099 | 0.855 (0.710 to −1.030) |
| Alcohol drinking | −0.054 | 0.100 | 0.289 | 0.591 | 0.948 (0.780–1.152) |
| Diabetes | 0.167 | 0.091 | 3.366 | 0.067 | 1.181 (0.989–1.411) |
| Hypertension | 0.313 | 0.077 | 16.525 | **<0.001** | 1.368 (1.176 to −1.591) |
| GLU | −0.004 | 0.013 | 0.091 | 0.763 | 0.996 (0.972 to −1.021) |
| TC | 0.020 | 0.057 | 0.130 | 0.719 | 1.021 (0.913 to −1.141) |
| BUN | 0.050 | 0.021 | 5.477 | **0.019** | 1.051 (1.008–1.096) |
| LDL-C | −0.101 | 0.070 | 2.050 | 0.152 | 0.904 (0.788–1.038) |
| Post-dilatation | 0.032 | 0.077 | 0.169 | 0.681 | 1.032 (0.888–1.199) |
| Number of vascular lesions | 0.185 | 0.091 | 4.142 | **0.042** | 1.203 (1.007–1.437) |
| CTO | 0.234 | 0.087 | 7.296 | **0.007** | 1.264 (1.066–1.498) |
| MLs | 0.011 | 0.164 | 0.004 | 0.948 | 1.011 (0.733–1.393) |
| Aspirin | −0.583 | 0.086 | 46.323 | **<0.001** | 0.558 (0.472–0.660) |
| RPP | 0.248 | 0.078 | 10.090 | **0.001** | 1.281 (1.099–1.492) |

The boldfaced values indicate p<0.05.
BUN, blood urea nitrogen; CTOs, chronic total occlusion lesions; GLU, glucose; LDL-C, low-density lipoprotein cholesterol; MACEs, major adverse cardiovascular events; MLs, multivessel lesions; RPP, rate pressure product; TC, total cholesterol.

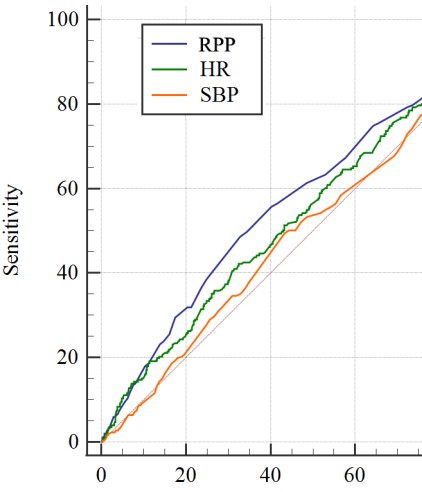

**Figure 3** ROC curves of RPP, HR and SBP and their respective area under the curve (demonstrating the ability of each parameter to predict the occurrence of cardiac death. HR, heart rate; ROC, receiver operating characteristic; RPP, rate pressure product; SBP, systolic blood pressure.

(HR=1.263, 95% CI: 1.092–1.462, p=0.002) and MACEs (HR=1.281, 95% CI: 1.099– 1.492, p=0.001) that occurred in the high-value group were still significantly higher than those in the low-value group. In the patients with stable CAD, after performing an adjustment of confounders, the ACM, CM, MACCEs and MACEs remained significantly different; while in the ACS group, there was no significant difference in adverse outcomes after adjusting for confounders (online supplemental tables 5–8).

Taking cardiac death as an example, figure 3 shows the ROC curves of RPP as well as heart rate and SBP for patients with CAD with cardiac death. The area under the curve (AUC) of RPP (AUC 0.586) was larger compared with heart rate (AUC 0.554) or SBP (AUC 0.511) (online supplemental table 9).

## DISCUSSION

Among patients with stable CAD who underwent PCI, higher preoperative RPP value led to 60.5%, 73.3%, 27.1%, and 31.5% higher relative risks in ACM, CM, MACCEs and MACEs, respectively. However, there was no significant correlation between the RPP and the adverse outcomes in patients with ACS. Compared with heart rate and BP alone, RPP has a higher predictive value for adverse clinical outcomes.

RPP is a very reliable indicator of myocardial oxygen uptake ($MVO_2$). $MVO_2$ depends on SBP, heart rate and ventricular wall tension. RPP is often used to evaluate workload and exercise load, and an increase in RPP indicates that subjects need to increase $MVO_2$ to meet the metabolic needs of the body.[16 17] RPP in healthy normotensive young adults has a normative circadian pattern, and the RPP value is greater during the day than at night, especially in the afternoon. The circadian pattern of the large amplitude in the RPP and its sex-based differences must be taken into account when using the RPP to assess

cardiac workload, risk of left ventricular hypertrophy and efficiency of antihypertensive therapy.[18] Stegehuis et al[19] performed intracoronary Doppler flow velocity measurements to obtain coronary flow reserve (CFR) and coronary flow capacity (CFC) after inducing hyperaemia in 390 non-obstructed vessels of patients who were scheduled for elective PCI of another vessel. Akaike's Information Criterion revealed that the RPP is an independent predictor of CFR and CFC. Since CFR and CFC are physiological indices for the assessment of myocardial blood flow damage caused by CAD, we can clarify the correlation between the RPP and the prognosis of patients with CAD who underwent PCI.

Furthermore, pathophysiological studies indicated that a relatively high heart rate has direct detrimental effects on the progression of coronary atherosclerosis, on the occurrence of myocardial ischaemia and ventricular arrhythmias, and on left ventricular function.[20] Previous studies have demonstrated that elevated heart rate was an independent risk factor for long-term adverse prognosis in patients with CAD after PCI who had postoperative stable angina or ACS.[21–23] O'Brien et al[24] observed a correlation between the heart rate and the prognosis of patients with CAD who underwent PCI and found that heart rate before PCI was an independent predictor of adverse 30-day cardiovascular outcomes. In addition, BP and pulse pressure (PP) were well-known independent cardiovascular risk factors. Compared with brachial PP, aortic PP was better to evaluate the extent of CAD.[25] SBP was found to be a risk factor for cardiac death in patients with CAD. BP before PCI was associated with long-term prognosis in patients with CAD, and the long-term mortality rate of patients with CAD with low systolic and high throbbing pressure was significantly reduced. In other words, a wider PP could be used as a risk factor for patients with CAD who underwent PCI.[26–28] A previous study indicated that 'the oscillatory gap', which was defined by the difference between the oscillatory SBP and the auscultatory SBP, can be used to predict the presence of CAD.[29] RPP combined the information of heart rate and SBP, and both the indicators were related to the prognosis of patients with CAD. Compared with SBP, DBP, average arterial pressure, heart rate and other indicators, RPP may have a higher predictive value,[12] which was consistent with our study.

At present, most studies mainly took RPP as one of the important evaluation indicators of disease-related rehabilitation training, which provided patients with reasonable exercise training guidelines for rehabilitation.[30 31] Moore et al[32] found that the overall risk of cancer mortality increased significantly in patients with no physical activity but with a higher RPP. In this study, we found that the patients with CAD who underwent PCI in the high-value group had higher incidence of mortality, MACCEs and MACEs.

There are many advantages. The first is the larger sample size, which can improve the scientific nature of the statistics. The second is the long follow-up time, which

can further improve the reliability of the statistical results. The third is that RPP combines the information of two indicators and is simple and easy to obtain. However, its limitations cannot be ignored. First, we only collected heart rate and BP data at the first hospitalisation, thus lacking information on dynamic changes in these variables. Second, the present study did not collect the data of left ventricular mass index and not rule out the effect of the patient's current medication such as antihypertensive drugs on BP levels, which may have impacted the results. Third, this study had a single-centre retrospective cohort design, and our results need to be further validated through large multicentre randomised controlled experiments. Moreover, follow-up was mainly based on telephone contacts, as well as the method of measurement was mainly manual, which may have biased the events. Finally, the linearity cannot be fully addressed for covariates in the Cox model.

There is a problem worth exploring. Xu et al[33] found that high RPP was associated with MACEs for patients with ACS who underwent PCI. However, the predictive value of the RPP weakened when adjusting for heart rate. As a result, they have shown that RPP may reflect the predictive power of heart rate for patients with ACS who underwent PCI, which was inconsistent with our study. This may be because the study involved a follow-up of only 2 years, and the study population and results were different; in particular, patients with cardiogenic shock and inpatients with MACEs in that study were excluded.

## CONCLUSION

In summary, RPP is an independent predictor of long-term prognosis in patients with CAD who underwent PCI. Therefore, the risk of disease can be stratified according to the RPP before PCI, which provides a basis for the treatment strategy of patients with CAD, and a new concept for preventing the occurrence of clinical outcomes.

**Acknowledgements** The authors are grateful to the Department of Cardiology at the First Affiliated Hospital of Xinjiang Medical University for their support and expertise in conducting this study.

**Contributors** XX and Y-TM conceived and designed the research. Z-HJ performed the research. Z-HJ, AA, T-TW and Y-YZ analysed the data. XX and Y-TM did the quality control of the study and revision. Z-HJ wrote the paper. XX was responsible for the revision of the final version of the article. XX was responsible for the overall content as the guarantor.

**Funding** This work was funded by the National Natural Science Foundation of China (82170345).

**Competing interests** None declared.

**Patient and public involvement** Patients and/or the public were not involved in the design, or conduct, or reporting, or dissemination plans of this research.

**Patient consent for publication** Not required.

**Ethics approval** The CORFCHD-PCI Study complied with the Declaration of Helsinki and the protocol was approved by the Ethics Committee of the First Affiliated Hospital of Xinjiang Medical University. Since the study was designed retrospectively, the Ethics Committee considered that it was not necessary to obtain informed consent from the eligible patients.

**Provenance and peer review** Not commissioned; externally peer reviewed.

**Data availability statement** Data are available upon reasonable request.

**ORCID iD**
Xiang Xie http://orcid.org/0000-0002-0632-4626

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
