## [Reviewer comments · BMJ Open]

ARTICLE DETAILS

TITLE (PROVISIONAL)	Rate pressure product as a novel predictor of long-term adverse outcomes in patients after percutaneous coronary intervention: A retrospective cohort study
AUTHORS	Jiang, Zhi-Hui; Aierken, Abudula; Wu, Ting-Ting; Zheng, Ying-Ying; Ma, Yi-Tong; Xie, Xiang

VERSION 1 – REVIEW

REVIEWER	Ishii, Hideki Nagoya Univ
REVIEW RETURNED	15-Oct-2022

GENERAL COMMENTS	The authors evaluated the clinical values of RPP on the clinical outcomes of CAD patients undergoing PCI. There are several critical issues in the manuscript. Specific comments: 1. The rate of GBMT was quite low.2. How many patients were lost for follow-up? Were they excluded? If so, please indicate the number of all patients undergoing PCI in the institution during the study period.3. In the study, both CCS and ACS might be evaluated. In ACS cases, vital signs were unstable.4. PCI procedures themselves may greatly affect the results. However, there is no data.5. LVEF, LVMI, and so on are also important predictive values. However, there was no data.6. Please indicate the definition of RBP 1 and 2 in Figure 2.
---

REVIEWER	El Tahlawi, Mohammad Zagazig University, Cardiology
REVIEW RETURNED	11-Nov-2022

GENERAL COMMENTS	The authors study Rate pressure product as a novel predictor of long-term adverse outcomes in patients after percutaneous coronary intervention . The study is interesting and well written. However some concerns should be raised. Minor concerns: 1-P5 L12: "In the end" better to be " at the end".
--

	2-In the section of "Data collection", you should not put "The purpose of the study" however this aim of the study should be added just after the introduction. 3-Please define RPP (The core of your study) in the introduction section. Major concerns: 1-The method of detection SBP and its measurement should be added in "methods" section in some details because this is the core of your study. This means the method of measurement, the device used "manual or automated" and its type....etc. 2-Please explain why you did consider this figure "10.269" as a cut-off for RPP in dividing your patient groups. 3-In table 4, Hypertension was insignificant predictor for ACM while RPP (which is the product of SBP and HR) was significant. How do you explain this result? The same was for CM in table 5. 4-I think you should add a paragraph in the discussion section discussing the relationship between hypertension parameters and CAD. I recommend this refernce "El Tahlawi, M., Abdelbaset, M., Gouda, M. et al. Can we predict the presence of coronary lesions from blood pressure measurement? A new clinical method. Hypertens Res 38, 260–263 (2015). & "François Philippe, Elie Chemaly, Jacques Blacher, Jean-Jacques Mourad, Alain Dibie, Fabrice Larrazet, François Laborde, Michel E. Safar, Aortic pulse pressure and extent of coronary artery disease in percutaneous transluminal coronary angioplasty candidates, American Journal of Hypertension, Volume 15, Issue 8, August 2002, Pages 672–677" 5- The limitations should be clarified including the reproducibility of such parameter, the single patient race examined, the method of measurement...etc.
--	--

REVIEWER	Wu, Yangfeng Peking University School of Public Health, Epidemiology
REVIEW RETURNED	13-Nov-2022

GENERAL COMMENTS	Using the product of heart rate and blood pressure to predict the long term clinical outcomes among patients after PCI sounds novel, but to establish the novelty it requires additional work to prove if this new index/vriable has value additional to heart rate and blood pressure in predicting the risks. Theoritically, since the heart rate and blood pressure are associated with the risk of clinical outcomes, their product should be certainly associated with the risks too. To propose the product as a novel predictor, its role independent to all traditional predictors should be proved. Otherwise, the conclusion can not be accepted.
--

VERSION 1 – AUTHOR RESPONSE

Reviewer:1

Dr. Hideki Ishii, Nagoya Univ

1. The rate of GBMT was quite low.

Reply: Thanks for the reviewer’s comments. We are very sorry that we could not understand what “GBMT” means. If you need further modification, please feel free to contact us.

2. How many patients were lost for follow-up? Were they excluded? If so, please indicate the number of all patients undergoing PCI in the institution during the study period.

Reply: Thanks for the reviewer's comments. In our study, regarding the inclusion criteria, coronary angiography confirmed that at least one major coronary artery showed stenosis $\geq 70\%$, including the left main coronary artery, left anterior descending artery, left circumflex artery, and right coronary artery, or other more important branches. All patients underwent PCI and received at least one stent via implantation. Regarding the exclusion criteria, patients with the presence of serious heart failure, rheumatic heart disease, valvular heart disease, congenital heart disease, pulmonary heart disease, serious dysfunction of the liver or kidney, acute infections, malignant tumor, alcohol abuse or blood disease were excluded. To investigate the relation between the RPP and outcomes in CAD patients who underwent PCI, a total of 6,050 patients were evaluated initially according to the inclusion and exclusion criteria. Thirty-five patients were excluded due to incomplete follow-up data. Finally, 6,015 patients were enrolled in this study, including 2,029 ACS patients and 3,986 stable CAD patients. All the 6,015 patients strictly met the inclusion and exclusion criteria, and the clinical data of them were complete. We have made changes in the revised version.

3. In the study, both CCS and ACS might be evaluated. In ACS cases, vital signs were unstable.

Reply: Thanks for the reviewer's comments. In the revised version, we supplemented the sub-group analysis to investigate the relationship between the RPP and the adverse outcomes in different sub-group. Details and forms are in the attachment and the main document.

4. PCI procedures themselves may greatly affect the results. However, there is no data.

Reply: Thanks for the reviewer's comments. We conducted a supplementary analysis of baseline treatments and procedure characteristics in accordance with the reviewers' suggestion. In the revised version, as shown in Table 2 above, several variables were significantly different between these two groups, especially in the ACS patients. These variables included post-dilatation pressure, chronic total occlusion lesions (CTO), multi-vessel lesions (ML) as well as the number of vascular lesions (all $P < 0.05$). We did not find significant difference in regards to new generation stent, diameter of stents, length of stents, the number of stents, the rate of pre-dilatation or post-dilatation (all $P > 0.05$).

5. LVEF, LVMI, and so on are also important predictive values. However, there was no data.

Reply: Thanks for the reviewer's comments. We conducted a supplementary analysis of LVEF in accordance with the reviewers' suggestion. As shown in the revised version, we did not find significant difference between the two groups in LVEF ($P = 0.65$). We did not collect the data of LVMI in our study, therefore, we cannot compare the performance between LVMI and RPP. We believed this is a limitation in our study and declared this limitation in the revised version.

6. Please indicate the definition of RPP 1 and 2 in Figure 2.

Reply: Thanks for the reviewer's comments. In this study, $RPP < 10,269$ was considered the low-value group, and $RPP \geq 10,269$ was considered the high-value group. Group 1 is the low-value group, and group 2 is the high-value group. We have supplemented the definitions of group 1 and 2 in Figure 2.

Reviewer: 2

Dr. Mohammad El Tahlawi, Zagazig University

Minor concerns:

1. P5 L12: "In the end" better to be " at the end".

Reply: Thanks for the reviewer's comments. We are very sorry for the error in the original version. I have corrected it in the revised version.

2. In the section of "Data collection", you should not put "The purpose of the study" however this aim of the study should be added just after the introduction.

Reply: Thanks for the reviewer's comments. We are very sorry for the error in the original version. I have corrected it in the revised version.

3. Please define RPP (The core of your study) in the introduction section.

Reply: Thanks for the reviewer's comments. The RPP was defined as the product of SBP (mmHg) and HR (beats per minute). We have added the above content in the revised version.

Major concerns:

1. The method of detection SBP and its measurement should be added in "methods" section in some details because this is the core of your study. This means the method of measurement, the device used "manual or automated" and its type....etc.

Reply: Thanks for the reviewer's comments. HR and blood pressure (BP) at admission were measured accurately according to a highly standardized protocol at the first medical contact by experienced physicians using calibrated manual mercury sphygmomanometers with an appropriately sized cuff. One sphygmomanometer was placed on the right arm, and after at least 5 min of rest in the sitting position. Two HR and BP measurements were obtained and taking the average, allowing for a 1 min interval between measurements. In the revised version, we have added the above content in "methods".

2. Please explain why you did consider this figure "10.269" as a cut-off for RPP in dividing your patient groups.

Reply: Thanks for the reviewer's comments. In our study, because there is no established optimal threshold for RPP, receiver operating characteristic (ROC) curve analysis was performed to assess the predictive ability of RPP in clinical outcomes. The ROC curve analysis showed that a RPP value of 10,269 was the cut-off with the highest sensitivity and specificity in terms of prognostic significance. The population clinical characteristics were grouped by dichotomy of RPP, which was based on the ROC cut-off. $RPP < 10,269$ ($n=4,018$) was considered the low-value group, and $RPP \geq 10,269$ ($n=1,997$) was considered the high-value group.

3. In table 4, Hypertension was insignificant predictor for ACM while RPP (which is the product of SBP and HR) was significant. How do you explain this result? The same was for CM in table 5.

Reply: Thanks for the reviewer's comments. Hypertension was diagnosed when SBP was ≥ 140 mmHg and/or diastolic blood pressure (DBP) was ≥ 90 mmHg following repeated examination, or treatment with antihypertensive drugs [14]. Because hypertension was not diagnosed by SBP alone, and RPP is the product of SBP and HR, the effects of hypertension and RPP on ACM and CM can be different.

4. I think you should add a paragraph in the discussion section discussing the relationship between hypertension parameters and CAD. I recommend this reference "El Tahlawi, M., Abdelbaset, M., Gouda, M. et al. Can we predict the presence of coronary lesions from blood pressure measurement? A new clinical method. *Hypertens Res* 38, 260–263 (2015). & "François Philippe, Elie Chemaly, Jacques Blacher, Jean-Jacques Mourad, Alain Dibie, Fabrice Larrazet, François Laborde, Michel E. Safar, Aortic pulse pressure and extent of coronary artery disease in percutaneous transluminal coronary angioplasty candidates, *American Journal of Hypertension*, Volume 15, Issue 8, August 2002, Pages 672–677".

Reply: Thanks for the reviewer's comments. In the revised version, the above papers have been added to the discussion. In addition, BP and pulse pressure (PP) were well-known independent cardiovascular risk factors. Compared brachial PP, aortic PP was better to evaluate the extent of coronary artery disease [25]. A previous study indicated that "the oscillatory gap", which was defined by the difference between the oscillatory systolic blood pressure (OSBP) and the auscultatory systolic blood pressure (AUSBP), can be used to predict the presence of CAD [29].

5. The limitations should be clarified including the reproducibility of such parameter, the single patient race examined, the method of measurement...etc.

Reply: Thanks for the reviewer's comments. We have discussed the limitations of the research more extensively in the revised version. First, we only collected heart rate and blood pressure data at the first hospitalization, thus lacking information on dynamic changes in these variables. Second, the present study did not collect the data of LVMI (left ventricular mass index) and not rule out the effect of the patient's current medication such as antihypertensive drugs on blood pressure levels, which may have impacted the results. Third, this study had a single-centre retrospective cohort design, and our results need to be further validated through large multi-centre randomized controlled experiments. And follow-up was mainly based on telephone contacts, as well as the method of measurement was mainly based on manual, which may have biased the events. Finally, the linearity cannot be fully addressed for covariates in the Cox model. However, for the patient race examined, since Xinjiang is an area inhabited by multi-ethnic groups, the limitation of the patient race examined would not be considered for the time being.

Reviewer:3

Dr. Yangfeng Wu, Peking University School of Public Health

Using the product of heart rate and blood pressure to predict the long term clinical outcomes among patients after PCI sounds novel, but to establish the novelty it requires additional work to prove if this new index/variable has value additional to heart rate and blood pressure in predicting the risks. Theoretically, since the heart rate and blood pressure are associated with the risk of clinical outcomes, their product should be certainly associated with the risks too. To propose the product as a novel predictor, its role independent to all traditional predictors should be proved. Otherwise, the conclusion can not be accepted.

Reply: Thanks for the reviewer's comments. We conducted a supplementary analysis in accordance with the reviewers' suggestion. In the revised version, Figure 3 shows the ROC curves of RPP as well as HR and SBP for CAD patients with cardiac death. The results of the ROC curve analysis for different parameters are given in Table 9. The AUC of RPP (AUC 0.586) was larger compared to HR (AUC 0.554) or to SBP (AUC 0.511). In addition, we also carried out supplementary analysis in COX regression analysis. Details and forms are in the attachment and the main document.

VERSION 2 – REVIEW

REVIEWER	Ishii, Hideki Nagoya Univ
REVIEW RETURNED	22-Jan-2023

GENERAL COMMENTS	The reviewer found that the revised manuscript has been significantly improved. However, there are a few issues which should be modified. 1. GBMT means guideline-based medical therapy. Data on some medical treatments are shown in Tables. However, the rate of GBMB seems relatively low for the current guidelines. Data on anti-platelet agents are essential. 2. Infarction size was not considered in patients with AMI.
---

REVIEWER	El Tahlawi, Mohammad Zagazig University, Cardiology
REVIEW RETURNED	20-Jan-2023

GENERAL COMMENTS	Satisfactory response. Thank you
----------------------------------

VERSION 2 – AUTHOR RESPONSE

Reviewer: 2

Dr. Mohammad El Tahlawi, Zagazig University

Comments to the Author:

Satisfactory response. Thank you

Reply: Thanks for the reviewer's comments.

Reviewer: 1

Dr. Hideki Ishii, Nagoya Univ

Comments to the Author:

The reviewer found that the revised manuscript has been significantly improved.

However, there are a few issues which should be modified.

Reply: Thanks for the reviewer's comments.

1. GBMT means guideline-based medical therapy.

Data on some medical treatments are shown in Tables. However, the rate of GBMT seems relatively low for the current guidelines.

Reply: Thanks for the reviewer's comments. As coronary artery disease is a long-term chronic disease with long medication duration, during which it is influenced by a variety of factors, leading to unauthorized discontinuation or reduction of dosage in most patients, which affects the final outcome of treatment. Several studies had shown that the CAD patients had good adherence to secondary prevention, but there was still room for improvement. The number of episodes, doctor-patient relationship, literacy, income, medical insurance, occurrence of drug side effects, and the knowledge of coronary artery disease were independent factors affecting patients' adherence to medication. In our study, the data of medical treatments we collected were not the treatment regimens at discharge, but the current medications used at the time of follow-up. The CAD patients who underwent PCI had formal medication regimens at discharge, but adherence declined over time, with patients adjusting their medications on their own. Xinjiang was a multi-ethnic region with a complex demographic structure; patient compliance was declining year by year; and our study was a 10-year follow-up study, all of which were reasons why the rate of GBMT seemed relatively low.

Data on anti-platelet agents are essential.

Reply: Thanks for the reviewer's comments. We conducted a supplementary analysis of aspirin and clopidogrel in accordance with the reviewers' suggestion. As shown in the revised version, we find significant difference between the two groups in aspirin ($P < 0.001$), the difference being mainly in stable CAD patients. But we did not find significant difference between the two groups in clopidogrel ($P = 0.626$).

2. Infarction size was not considered in patients with AMI.

Reply: Thanks for the reviewer's comments. Data on the number of vascular lesions was shown in Table 2. The number of vascular lesions was a lateral response to the infarction size, which was not measured during the PCI.